# Proteomics—The State of the Field: The Definition and Analysis of Proteomes Should Be Based in Reality, Not Convenience 

**DOI:** 10.3390/proteomes12020014

**Published:** 2024-04-19

**Authors:** Jens R. Coorssen, Matthew P. Padula

**Affiliations:** 1Department of Biological Sciences, Faculty of Mathematics and Science, Brock University, St. Catharines, ON L2S 3A1, Canada; 2Institute for Globally Distributed Open Research and Education (IGDORE), St. Catharines, ON L2N 4X2, Canada; 3School of Life Sciences and Proteomics, Lipidomics and Metabolomics Core Facility, Faculty of Science, University of Technology Sydney, Sydney, NSW 2007, Australia

**Keywords:** integrative top-down proteomics, bottom-up proteomics, immunoassay, mass spectrometry, protein species, proteoforms, structural prediction, tandem MS, two-dimensional gel electrophoresis, Western blotting, multi-omics, isoelectric focussing

## Abstract

With growing recognition and acknowledgement of the genuine complexity of proteomes, we are finally entering the post-proteogenomic era. Routine assessment of proteomes as inferred correlates of gene sequences (i.e., canonical ‘proteins’) cannot provide the necessary critical analysis of systems-level biology that is needed to understand underlying molecular mechanisms and pathways or identify the most selective biomarkers and therapeutic targets. These critical requirements demand the analysis of proteomes at the level of proteoforms/protein species, the actual active molecular players. Currently, only highly refined integrated or integrative top-down proteomics (iTDP) enables the analytical depth necessary to provide routine, comprehensive, and quantitative proteome assessments across the widest range of proteoforms inherent to native systems. Here we provide a broad perspective of the field, taking in historical and current realities, to establish a more balanced understanding of where the field has come from (in particular during the ten years since Proteomes was launched), current issues, and how things likely need to proceed if necessary deep proteome analyses are to succeed. We base this in our firm belief that the best proteomic analyses reflect, as closely as possible, the native sample at the moment of sampling. We also seek to emphasise that this and future analytical approaches are likely best based on the broad recognition and exploitation of the complementarity of currently successful approaches. This also emphasises the need to continuously evaluate and further optimize established approaches, to avoid complacency in thinking and expectations but also to promote the critical and careful development and introduction of new approaches, most notably those that address proteoforms. Above all, we wish to emphasise that a rigorous focus on analytical quality must override current thinking that largely values analytical speed; the latter would certainly be nice, if only proteoforms could thus be effectively, routinely, and quantitatively assessed. Alas, proteomes are composed of proteoforms, not molecular species that can be amplified or that directly mirror genes (i.e., ‘canonical’). The problem is hard, and we must accept and address it as such, but the payoff in playing this longer game of rigorous deep proteome analyses is the promise of far more selective biomarkers, drug targets, and truly personalised or even individualised medicine.

## Abbreviations

2DETwo-dimensional gel electrophoresisIEFIsoelectric focusingSDS-PAGESodium dodecyl sulphate-polyacrylamide gel electrophoresisMWMolecular weightMSMass spectrometryLCLiquid chromatographyTMSTandem mass spectrometryTDPTop-down proteomicsiTDPIntegrative top-down proteomicsMSi-TDPMass spectrometry intensive top-down proteomicsBUBottom upBUPBottom-up proteomicsORFOpen reading framePTMPost translational modificationFTICR-MSFourier transform ion cyclotron resonance mass spectrometryLFQLabel-free quantificationpIIsoelectric point

“We choose to go to the Moon in this decade and do the other things, not because they are easy, *but because they are hard*.” John F. Kennedy; address at Rice University, Houston, Texas, 12 September 1962 (our italics).

## 1. Introduction

With origins most logically traced to the development of two-dimensional gel electrophoresis ((2DE) combining isoelectric focusing (IEF) and SDS-PAGE in progressive dimensions of separation) which enabled the resolution of hundreds (likely many thousands) of proteoforms/protein species [1,2,3,4,5,6,7,8,9,10], proteomics has undergone notable changes. Along with other versions of gel-based 2D separations (i.e., chromatography in a gel matrix), changes have included almost five decades of 2DE optimization yielding a truly high-resolution analytical platform for proteoforms across very broad ranges of charge (pI) and molecular weight (MW). Despite the development of new technologies in the intervening decades, *no other methods* provide such capacity for genuinely deep, comprehensive proteome analysis at the essential level of proteoforms [7,10,11]. With the introduction of mass spectrometry (MS) and particularly its coupling to liquid chromatography (LC), the coupling with tandem MS (LC/TMS) proving most productive at scale [12,13,14,15], proteomics has become a discipline in its own right. This combination of high-resolution technologies—2DE/LC/TMS—fully enables the highest resolution analytical chemistry approach to proteome analysis that is now most widely and appropriately referred to as top-down proteomics (TDP) [7,8,9,10,16,17].

Piggybacking on developing gene and amino acid sequencing methods, and the then pending and subsequent first release of the human genome, an alternate—bottom-up (BU) or ‘shotgun’—approach to proteome analysis came into vogue [18,19,20]. Notably, in the 1990′s, gene and protein sequencing faced similar problems in that methods were slow and increasingly unreliable with larger molecular sizes. For genomics, this led to short-read DNA sequencing in which 100–200 ’reads’ ensure accuracy. BU proteomics (BUP) has sought increased speed of canonical protein identifications (i.e., linkage to recognized/canonical gene sequences) rather than analytical depth at the critical level of proteoforms. Thus, this purely proteogenomic approach relies on gross digestion of complex native proteome extracts, LC for rudimentary sorting of the resulting peptide milieu (i.e., seeking to reduce the resulting complexity), TMS to sequence individual peptides, and software to link these to protein sequences or predicted Open Reading Frames (ORFs) in databases. Much of this is said to be undertaken in a relatively unsupervised manner, and purportedly with less hands-on technique relative to 2DE/TMS approaches. Notably, the replicate determinations recognised as critical in DNA sequencing never seem to have been viewed with the same rigour by most BUP practitioners (but are fortunately standard practice in most 2DE/TMS analyses) [7,8,11]. Shortcomings and the need for more routine rigour in BUP have only become slowly apparent, as is not unusual with new research approaches [7,21,22,23,24,25,26,27,28,29,30]. Nonetheless, while initially providing some ease in terms of analytical methodology, and reasonable coverage of high abundance canonical amino acid sequences, the vast majority of BUP identifications are inferences based often on only one or a couple of peptides (i.e., not even approximating the full length of the amino acid sequence in databases). Highly abundant sequences tend to dominate the findings. However, beyond the widely recognized problems of inferring canonical protein identifications and assuming the presence of intact/full length species, the most serious issue by far is the loss of all information concerning *specific* proteoforms. Indeed, early work with this approach even ‘corrected’ datasets to eliminate isoforms and PTM in order to simplify database searching and focus only on canonical amino acid sequences. This is another example of supposed simplification obscuring rather than addressing proteome complexity. Unfortunately, recent BUP representations of proteoform ‘groups’ derived from the identification of modified peptides also do not address the critical issue of quantitatively identifying *specific* proteoforms [31]. This merely extends the inference problem by vaguely indicating that *some* proteoforms are apparently present (or at least a modified peptide is part of the peptidome). Essentially, this only moves us from an inferred ‘canonicome’ to the speciation of peptides. In the end, such inference, rather than definitive resolution and identification, is comparable to epidemiology in that it cannot establish definitive characteristics or causes (i.e., a specific proteoform) but only correlations (i.e., of a peptide to a sequence in a database). Nonetheless, should a case arise in which a specific proteoform is, for example, definitely linked to a function or disorder, then a targeted assessment of one or more of its distinctly characteristic (i.e., modified) peptides might serve for screening.

Therefore, we submit that BUP as currently and popularly applied is a quick cataloguing tool but, in and of itself, cannot provide the data needed for deep, critical, systems-level understanding of biological processes [10,32,33,34,35,36]. Identification pipelines that reject certain ‘protein’ identifications because, a decade or more ago, they were thought to be ‘contaminants’ or ‘routinely appearing’ [37] need to have their appropriateness reassessed. As proteoforms were never identified in the original studies using BUP identification approaches, one must consider that these studies and resulting databases ignore potentially critical proteoforms as the original analyses considered only canonical amino acid sequences correlated years ago with gene databases. It is, however, crucial to note that BUP is a powerful component of Integrated or Integrative TDP (iTDP) analytical approaches (see below).

## 2. Where Things Stand and Why

Realistically, we are now in the *post-proteogenomic era*, and likely (should) have been for well over a decade or more already. Change can be hard but only genuinely deep, comprehensive proteome analyses at the critical level of proteoforms will provide the data necessary to identify rational biomarkers and therapeutic targets via extensive dissection of molecular mechanisms and pathways. This critical approach will provide objective, systems-level understanding of biology, in particular coupled with a growing appreciation of genome complexity, gene regulation, epigenetics, and metabolomics. Indeed, a recent suggestion is that the field should be doing “proteoformics”—focusing on proteoforms—rather than proteomics [38]. The reader can draw the parallel to the JFK quote above, that researchers in proteomics need to focus on what is hard because it will ultimately provide the most critical, informative, and thus essential data. Continuing on the current ’easy’ path brings to mind Einstein’s (misattributed) quote that “Insanity is doing the same thing over and over and expecting different results” [39].

In this Perspective, we will take a hard look and ask some hard questions as it is high time for the field to come to grips with them. To understand how we intend to identify/address these issues, it might be best for readers to first consider the ‘what if’ questions relative to what is now finally acknowledged as the real complexity of proteomes [10,33,34,35,36,40,41,42,43,44]. What if the field had progressed differently? Think deeply and purely objectively for a moment about what has transpired and in terms of the present and future. What if the field had not taken a predominantly proteogenomic strategy over the last 20+ years? Minimally, we knew at the time that there were undoubtedly a substantial number of variants to any gene product or protein (e.g., mutations, alternate and multiple reading frames, splice variants, posttranslational modifications (PTM) [1,45]), and thus that BUP could never provide the depth of analysis needed for genuinely comprehensive proteome analyses [46,47,48]. Nonetheless, ‘fast’ and ‘easy’ analyses linking only to the genome became the standard. Some prominent journals even established BUP as their sole focus, rejecting any other studies, even when they employed higher resolution approaches. Notably, it was also already known that 2DE coupled with MS could resolve and identify many thousands of proteoforms from proteome extracts [2,49,50,51,52,53,54]. What if, rather than going down the purely proteogenomic rabbit hole, we had fully utilised the truly high resolution 2DE/LC/MS analyses—*a genuine, comprehensive and integrative TDP approach* [3,4,5,6,8,10,17,55]—how many (human) proteoforms would our databases already contain, minimally at the level of pI and MW if not actual PTM, along with the canonical amino acid sequence? How many more selective antibodies (and antibody therapeutics) would already be available? How many highly specific/selective biomarkers and drugs would already be available (or at least close to market) to address critical healthcare burdens? Even individualised medicine? We leave it to the concerned reader to provide a critical if only conservative estimate. A better, transparent, more collegial, complementary, and thus integrated path forward is clearly needed.

## 3. What Is Proteomics? What Is a Proteome? Defining Issues to Date

The first issue is recognition that proteomics deals exclusively in proteoform abundance, *not* protein expression or up/down-regulation. The latter require different assays. Even when correlations with mRNA levels happen to exist, these do not fully establish that changes in canonical protein levels are due solely to changes in gene expression unless stability of the species (e.g., degradation rate) is also assessed. We consistently see such terms misused in the literature, particularly in studies not published in rigorous proteomics journals. In this regard, it is also frustrating to see references to a gene having or doing a certain function; genes are codes, not functional/active entities, and thus do not otherwise ‘do’ anything.

Appropriately defining a proteome is the next and more important issue that must be addressed in developing a truly comprehensive (i.e., deep and quantitative) and broadly unified analytical approach. Although the term “proteome” was first coined by Marc Wilkins and colleagues in 1995 [56], realisation of the genuine complexity of proteomes now demands that, beyond simply the canonical amino acid sequences encoded by the genome, proteomes be most accurately defined by their proteoform constituents. This should then be further refined by location/space (e.g., particular cells, subcellular compartments/organelles, molecular complexes) and time, as the proteome is highly dynamic relative to the genome and transcriptome. Despite these critical considerations, to ensure the highest quality proteome analyses, a core issue remains the criteria for defining a proteome.

Regrettably, in considering the literature over the last decade or more, ‘proteome’ seems largely to have become a term of convenience rather than rigour. To fully address the complexity of proteomes, the simplest first step is recognition that the word ‘proteoform’ should most appropriately replace the generic term ‘protein’ in almost all usage other than general references to that latter group of macromolecules; in the case of proteogenomic/BUP studies, the data are most appropriately referred to as ‘inferred ORF products’ [45]. Perhaps it would help if that change was made in Wikipedia so that, earlier in their education, students already understand and accept the real complexity inherent to proteomes? Simply continuing to introduce students to the Central Dogma in high school is painfully outdated and insufficient for their future endeavours, or for research efforts overall.

However, rather than such a critical and objective approach, defining the proteome seems to have become a method-dependent matter of convenience. For BUP studies, this largely means inferred identification and quantification of apparent proteogenomic/canonical ORF products without mention of the lack of isoform or proteoform identifications or the problems of protein inference. For MS-intensive TDP (MSi-TDP) [7,57,58], this essentially means some sub-proteome, largely if not solely within the <20–30 kDa size range of total species in the proteome [59]; while these methods are very occasionally able to analyse a higher MW species, this amounts to a vanishingly low fraction of any given proteome [60]. Although some published studies refer to these as ‘comprehensive’ proteome analyses, this is at best misleading. While MSi-TDP can provide incredibly detailed assessments of proteoforms and isolated complexes that conform to the capabilities of the method, these are not quantitatively or even qualitatively deep, comprehensive assessments of proteomes (i.e., across a breadth of pI and MW that defines native proteoforms in cells, tissues, and biological fluids). Is this, and the very expensive, high-end MS instrumentation required (e.g., FTICR MS), a promising approach? Absolutely. But, there have been limited advances in this approach for the last 1–2 decades due largely to the diversity and dynamic range of the proteome (e.g., notably mid-to-large MW and membrane species), poor front-end resolution of species even when combining multiple separation steps (i.e., still resulting in co-elution of small and large species), the decay in signal-to-noise with large species due to a plethora of charge states, and the need for better software integration, as well as even more effective dissociation methods to fragment larger species [58,61,62,63,64]. Indeed, while smaller proteins have been analysed by MS since the early 1960s, including with early software programs [65,66,67,68,69,70], some of the first reports of mid-to-large proteins being analysed by MSi-TDP were from the early 1990s until the mid-2000s [58,71,72,73,74]. Thus, there has been little substantive advance in terms of a broad and consistent breaking of the current ~20–30 kDa ‘MW barrier’. Routine full proteome analyses using the MSi-TDP approach must still await further developments and rigorous testing. Perhaps its strongest immediate application is in the analysis of the small proteoforms inherent to isolated proteins and protein complexes [75,76,77,78,79,80]. In this regard, it is also important to note recent advances in MSi-TDP instrumentation that enable effective analysis of low MW species by capitalising on the practical mass resolution of instruments used for BUP, these being able to resolve the isotopic series of proteoform charge state (e.g., Exploris 120/240/480, TIMSTOFs, ZenoTOFs) (e.g., [81]). Accordingly, MSi-TDP can and should be used by BUP practitioners as a complementary technique enabling a more critical analysis of at least a fraction of the proteome. Furthermore, considering inherent issues with the front-end protein separation methods routinely used in MSi-TDP—e.g., the misleadingly acronymed Gel-Eluted Liquid Fraction Entrapment Electrophoresis (GELFrEE), a 1D separation utilising tube gels and SDS which must be removed prior to LC or MS [82]—employing 2DE, which, by design, ‘isolates’ proteoforms during the two-step resolving process, would likely promote deeper analyses, perhaps even of larger species. To date, such a critical approach remains untested [83]. MSi-TDP practitioners continue to use combinations of GELFrEE and/or multiple LC phases, despite recognised issues of co-eluting large and small species and complex spectra that require multiple software tools for downstream analyses that can take multiple hours or even longer to complete yet can still yield ambiguous identifications.

Thus, rather than methods-centric, methods-dependent, or otherwise insular definitions, the simplest, most objective and straightforward definition of a proteome is that it is a specific collection of proteoforms that are intrinsic to the native state at the time of sampling. What that native state is must be specifically defined: sample source/type, including specific (sub)fractions; all details of sample handling/processing and any ‘fractionation’; all details of downstream analysis including specifics of any and all resolving protocols (e.g., gel and/or LC, and MS), data processing, and final broad availability of the data. The latter has, of late, become an interesting concern as genome and protein analyses and databases become more refined, leading to the question of how many canonical protein identifications (e.g., from a decade or more ago) would still be accurate if reassessed? Perhaps there are already critical data in the literature that have been ‘missed’ and, likewise, red herrings misleadingly identified as critical that should not have been a focus had better data/interpretation been the original outcome. Therein lies the critical need for more complete amino acid sequence coverage and resolution of proteoforms [84,85,86,87,88,89], constant refinement of statistical and bioinformatics tools, and data banking [10]. Furthermore, raw gel images, LC chromatograms, and MS data for all published studies must be made available on publicly accessible data repositories. While this is standard rigour at all critical proteomics journals, many medical and other journals still do not demand this assurance of data reproducibility, often enabling the publication of proteomic research of questionable quality. In summary, any study referred to as unbiased, global, (ultra)deep or otherwise ‘comprehensive’ when it does not address proteoforms but only canonical amino acid sequences and/or ‘sub-proteomes’ within only limited MW ranges, or that does not fully establish its reproducibility, is quite unrealistic and disingenuous.

Fundamentally, it is only the ‘detectable’ proteome that is defined by the methodologies employed. To address limitations of methods/instrumentation, it has become common practice to reduce sample complexity, with all the inherent risks that entails. To reduce data complexity and aid ease of interpretation, the sample is fractionated according to physicochemical properties (e.g., proteoform size and/or charge, peptide hydrophobicity, complex size), either to focus on a compartment (e.g., nucleus, membrane, vesicle) or increase identification depth (qualitatively, sometimes quantitatively). In this vein, PTM characterisation using BUP currently favours enrichment of the modified peptides in an effort to overcome their generally lower abundance in the total peptide milieu compared to the unmodified peptides. Such approaches not only disconnect PTM from the specific proteoform (see above) but also bias research toward PTM that have purportedly reasonable enrichment strategies. This is why phosphorylation is the most studied PTM. Therefore, how do multi-step fractionation and enrichment protocols affect sample quality (e.g., proteoform/PTM lability) and thus the qualitative and quantitative accuracy of analyses? Again, the concept of proteoform groups, based on peptide speciation in BUP, is a methods-limited extension of the inherent limitations of inference. In the end, is there really any 100% suitable solution to truly comprehensive proteome analysis aside from analysing the native sample as close to its native state as possible?

## 4. Recognising and Addressing Critical Issues

To begin, we should highlight progress made in the last ~30 years, using the first report to correlate peptide data with canonical amino acid sequences in databases [18] as a benchmark, and progress made in the 10 years of the journal Proteomes existence. There is certainly no debate that inference of canonical ‘protein’ identities based on peptide data from a shotgun analysis has become the most widespread approach in proteomics. Although BUP has been most concerned with a suggested speed or ease of analysis and correlations with canonical gene sequences (i.e., proteogenomics) [90], it has to some extent also sought to be quantitative where possible. Thus, label-free quantification (LFQ), while clearly having some notable limitations [91,92,93,94], including the inherent failure to discern peptide distributions between proteoforms [32,95] and the missing values problem [96,97,98,99], can prove reasonably informative when used judiciously [48]. Similarly, a range of peptide labelling techniques (e.g., Tandem Mass Tags) have sought to routinely quantify changes in the abundance of canonical proteins, although there are quite critical concerns to be taken into account in employing any labelling methods [10,100,101]. What then are some of the most critical concerns and advances in proteome analysis to date and, respectively, how must these be addressed and further developed and optimised to ensure genuinely deep, quantitative proteome analysis at the level of proteoforms?

### 4.1. Improvements in Proteoform Extraction and Sample Processing

Numerous concerns arise from the outset of sampling [102,103,104]. We base this on our firm belief (or mantra) that *the best proteomic analyses reflect, as closely as possible, the native sample at the moment of sampling*. Although desirable to immediately process the sample for analysis, this is clearly not feasible in most studies. Some studies take care to sample and store as quickly as possible (e.g., immediately snap freezing in liquid nitrogen and storing at −80 °C), while others likely incur artefacts by prolonged handling or further processing at room temperature (or higher), either pre- or post-freezing, and/or simply slow freeze the sample by placing it directly at −30 °C or −80 °C. While this problem may well be sensitive to sample volume, it appears to concern most research on blood serum and plasma. These are, in fact, far more problematic in terms of sampling, which will vary according to the gauge of needle used during phlebotomy, the length of time and the temperature at which the whole blood samples are kept prior to and during processing, and which anticoagulant is used to collect plasma or how the blood is allowed to coagulate to collect serum. Taking all these critical factors into account, it is clear that, in many studies, inter- and intra-sample variability mean that neither serum nor plasma represent ‘native’ or well-controlled replicates [105,106,107,108]. Considering cell and platelet lysis or activation that occurs during phlebotomy and/or blood processing, one must seriously ask about the definitions of serum and plasma—are these meant to represent components free in the blood during circulation in vivo or to refer to the total complement of components in the blood, including the contents of cells and platelets? It has also become clear that what is lost to the red cell fraction must also be taken into account [109,110]. Again, from a proteomics (or any ‘omics) perspective, it is critical to know all details of sampling and processing. This becomes particularly relevant in searching for biomarkers or potential therapeutic targets.

Regarding sample extraction, testing new detergents, chaotropes, and surfactant combinations has been a critical focus in proteomics since the development of 2DE [111,112,113]. The challenge both for the first dimension of 2DE and for MS analyses has been the incompatibility of SDS, which is broadly considered the gold standard for proteoform extraction. Nonetheless, effective combinations of automated frozen disruption, detergents, and denaturing agents have enabled even the resolution of complete membrane proteomes by 2DE, despite long-held dogma suggesting this was not feasible [4,113,114,115,116,117,118,119,120,121]. Indeed, refined extraction and 2DE protocols have established quantitative proteome analysis consistent with SDS extraction [122]. Cleavable (e.g., acid-labile) or photodegradable surfactants have also been used in BUP and MSi-TDP but these do not appear to have been widely adopted despite extraction potentially comparable to SDS [123,124]; these may negatively impact PTM and do not help overcome the MW limitations of MSi-TDP analyses. Another new avenue includes the use of ionic liquids and other extraction media to improve recovery of otherwise insoluble proteoforms. In the case of ionic liquids, there is evidence that certain combinations cause artefactual modification of proteoforms through backbone cleavage or side chain modification [125], requiring further investigation of these otherwise promising extraction reagents. Overall, depending on the sample type and/or focus of the analysis, it would be prudent to ensure optimization of extraction conditions, in particular if total proteome analysis is the goal. It is unlikely that one size fits all when it comes to total extraction of proteoforms (or as close to it as is possible).

Evidence that sample reduction (to effectively remove disulfide bonds) has been largely underpowered in most studies to date, and that optimization of this critical step further enhances TDP analyses, again emphasises the need to continuously and rigorously evaluate even long-established protocols [126]. The caveat remains, however, that we can never be fully certain that we have quantitatively recovered every copy of every possible proteoform from any given sample or sample type, particularly when recovery steps (i.e., centrifugation) are used to remove notable ‘insoluble’ materials, which seems more often to represent less quantitatively rigorous methodology. This is also particularly true when any purification steps are used in an attempt to isolate specific cellular fractions or proteoforms [127]. In affinity or other fractionation methods, appropriately thorough analyses demand that both resulting fractions be analysed in order to establish and account for the quantitative capacity of the protocol [128]. This is only quite rarely seen in the proteomics literature but is most consistent with good analytical practice.

While we unfortunately continue to see studies that still fail to use any or only minimal protease inhibitors during sampling, the use of broad spectrum protease, kinase, and phosphatase inhibitors, while not an exhaustive approach, was introduced two decades ago in an effort to preserve the native state of proteoforms as best possible [114,129]. This practice should be extended to include other proven small molecule inhibitors of PTM reactions; notably, however, this will also increase the cost of analyses. In the long-term, though, can we afford to get this wrong in terms of identifying critical proteoforms? Similarly, many studies utilise processing protocols in which proteome extracts (or peptides) are maintained at room temperature or higher for an hour or more (e.g., during chemical labelling protocols) [130]. It seems quite unlikely that such samples effectively reflect the native state of the proteome at the time of sampling, particularly in terms of labile PTM. We know for example that extended incubations even at reduced temperatures result in proteome/proteoform alterations [131]. Overall, then, sample processing times and conditions are of concern. Focused analysis of this issue would be useful to characterise and quantify the extent of proteoform changes, in particular to PTM. Furthermore, consistency and sensitivity in assessing total protein concentration in samples is essential to ensure quantitative comparisons [132]. Accordingly, the practice of assaying before organic precipitation, for example, but assuming total and consistent recovery of species should not be continued. Normalisation must be to the total protein content of each sample to be analysed.

### 4.2. Improvements in Proteoform Resolution by 2DE

Relative to LC and MS, 2DE protocols and instrumentation have likely undergone comparable if not greater refinement and optimization over the last 30 years; this now enables the deepest proteome analyses currently available by providing high resolution separation of the proteoforms inherent to any sample [4,5,10,55,122]. Indeed, deep analyses have confirmed that resolved ‘spots’ on 2D gels—the macro ‘visible’ overlapping of staining signals from many resolved micro spots of separate species migrating to almost the same location—contain multiple proteoforms. This enables reasonable estimations that a refined and optimised iTDP approach can resolve and identify ≥1 M proteoforms across a large pI and MW range, including low abundance species [4,55]. *There is no other current or developing analytical approach that provides such routine or deep proteome analyses*. In contrast to BUP, it is also important to note that a single spot from a 2D gel is a relatively simple sample compared to a gross, whole proteome digest, and likely the reason for much higher routine sequence coverage of species by iTDP.

The commercial availability of quality-controlled isolated pH gradient (IPG) gel strips for IEF established a consistent and thus highly reproducible first dimension for proteoform resolution [133]. Some might suggest the same is true of commercially available SDS-PAGE gels for the second dimension, although consistency in self-casting is easily achievable (e.g., using multi-casting chambers), and this also enables critical fine-tuning of gel composition (i.e., % acrylamide or all important gradient gels) and detergent choice/combination to optimise proteoform resolution depending on the nature of the sample [114,128]. Regrettably, this is often overlooked in favour of the convenience of precast gels, despite their cost, limits of resolution, and resulting plastic waste from the cassettes.

Considering the unparalleled resolving power of 2DE, it is quite disappointing to find that many studies using this technique fail to report the pI of species of interest. This is critical to fully capitalise on the resolving capacity of highly refined 2DE protocols by calibrating both the first and second dimensions of separation using appropriate standards and reporting both the pI and MW of species analysed. Relative to the canonical values in databases (i.e., calculated purely based on the amino acid backbone), this is the most straightforward first confirmation that a species of interest is a specific proteoform [4,116,117,119,120,121,128,134,135,136,137]. Staining 2D gels with PTM-selective stains (e.g., phospho- and glyco-protein reagents) can also provide front-end confirmation of certain modifications and, considering that these reagents can often be used in conjunction with total proteoform detection (e.g., colloidal Coomassie Brilliant Blue (cCBB)), it behoves researchers to extract as much information per gel as possible [4,128,138,139]. Well-planned studies can also capitalise on third separations (i.e., 2DE/3DE) to further resolve species obscured by hyper-abundant spots, as well as those at pI extremes and the gel front, providing still deeper proteome coverage within a single experiment, and further ‘simplifying’ subsequent MS analyses [4,114,128].

Of most recent note, the overall 2DE process has also undergone a substantial increase in throughput. Micro-perforating (i.e., microneedling) IPG strips significantly reduces rehydration loading time, thus saving about a day in the overall 2DE process and establishing that ‘faster’ processes can also be ‘better’ without sacrificing the quality or depth of quantitative analyses [140]. While there have also been reports of faster second dimension separations (i.e., PAGE), these rely on higher voltages and (local) temperatures and thus likely result in proteoform artefacts. Resolving the first and second dimensions at lower temperatures would still appear to be the best available approach [4,114,141]. Subsequent to 2DE, an alternate gel fixation protocol, avoiding organic solvents, further minimises the likelihood of artefactual alterations to resolved proteoforms [142]. Thereafter, numerous stains are available to detect the total proteome [138,139]. Of these, the most significant recent innovation may well be the development of a high sensitivity (e.g., femto-to-attomole) detection protocol that uses cCBB as a near-infrared dye [143,144,145,146]. Together with the development of a deep-imaging protocol to overcome the signal saturating effects of highly abundant species, this combined optimised approach can detect and identify even low abundance proteoforms within the total proteome [147]. In part, this is also due to the development and commercial availability of (1) high-resolution imaging equipment supporting multiwavelength excitation and emission (e.g., Odyssey imager (Licor, Lincoln NB); Typhoon^TM^ FLA-9000 (GE Healthcare); Typhoon 5 Biomolecular Imager (GE Healthcare)); and (2) image analysis software enabling high resolution spot identification (i.e., signal above local background) and detailed quantitative analyses (e.g., Delta2D (DECODON)). There is also ongoing development of quantitative image analysis approaches that may lead to the critical extraction of still more/better data from 2D gels [148,149,150,151,152,153]. Furthermore, the immunoblotting of 2D gels, even after staining, provides the most direct approach to quickly identifying proteoforms [3,52,154,155,156,157,158,159,160,161,162,163,164,165], provided the antibodies used have been critically vetted (including to PTM [166]), and even then there are caveats to consider (e.g., blockade of antibody binding by a given PTM) [10]. Furthermore, a well-established immunoblotting method ensures detection sensitivities in the femto-to-attomole range [141,167]. Often ignored, however, is the need to consistently assess transfer efficiency in order to ensure truly quantitative assessments; this is the analytical equivalent of assessing both the eluate and the retentate in rigorous affinity analyses.

At the interface of BUP and iTDP approaches is the need to digest samples for subsequent peptide analysis. One of the notable advances in this area has been the recognition that strong digestion conditions (i.e., high protease concentrations and/or the standard 37 °C) result in loss of lower abundance species and/or the overwhelming of their MS signals by protease autolysis products. Reducing both protease concentrations and incubation temperature yields far more reliable data in the form of increased sequence coverage per species assessed [4,120,121,137,147,168,169,170]. Additionally, there have been several reports of (ultra)fast digestion approaches although none seem to have come into widespread use, perhaps raising questions of quantitative losses [169,170,171,172,173].

### 4.3. Improvements in Liquid Chromatography

Regardless of the improvements in the speed and sensitivity of mass spectrometers, it is still impossible to analyse a whole proteome without some form of fractionation [174,175]. In BUP, many peptides will be essentially isobaric in their m/z value, leading to co-isolation for fragmentation, and thus complicated MS/MS spectra containing fragments from multiple peptides from different proteoforms [176,177,178]. In electrospray ionisation (ESI), it is well established that samples that are too complex cause ion suppression and non-detection of some analytes [179]. Thus, LC will continue to be an essential part of proteome analysis, whether coupled to MS or not. The current focus of BUP practitioners is increased throughput to obtain large datasets with comparable sample numbers to next generation genomic sequencing (NGS), to increase statistical power if not actual deep proteome coverage [29,180,181].

Despite assertions by instrument manufacturers’ marketing departments, LC as used in proteomics is often the last realm of the tinkerer, with set ups varying greatly from platform to platform depending on the operator. Maximum sensitivity, especially for analysis of 2DE-resolved proteoforms and single cell analysis, still requires nanoflow chromatography (<1 uL/min) using in-house manufactured columns with integrated ESI emitters that can reveal column-to-column reproducibility issues if these are not carefully quality-controlled [182]. A vendor-promoted move to microflow LC (1–5 uL/min) has resulted in more robust and reproducible separations and peptide ionisation, and significantly reduced cycle times, but requires a proportional increase in sample load to overcome reduced ionisation efficiency at higher flow rates [183]. In MSi-TDP, the LC co-elution of proteoforms with multiple overlapping charge states reduces sensitivity as molecules of the same proteoform are spread across multiple charge states, yielding data that are challenging to interpret [60,184]. Capillary electrophoresis has the potential to completely resolve individual proteoforms from others, resulting in simplified MS spectra, but the incompatibility of solvents and buffers with ESI and the extremely small injection volumes required have restrained its routine implementation [185,186]. Additionally, CE cannot address the charge state loss in sensitivity and suffers from the same limitation as BUP and other MSi-TDP approaches in that any experimental replicates must be carried out sequentially as parallel replicates are not possible.

### 4.4. Improvements in Mass Spectrometry

Advances in instrumentation have all focused on the same outcome: increasing the dynamic range of concentration able to be analysed and accurately quantified through either an increase in scan speeds or further separation of LC co-eluting ions inside the MS. The most popular solution has been the application of Trapped Ion Mobility Spectrometry (TIMS), as applied in the Bruker TIMSToF platform where the TIMS device serves two purposes: (1) accumulation of ions of the same type to increase sensitivity; and (2) mobility-based separation of batches of these ions to enable another level of separation than that permitted by LC alone, thus reducing the diversity of m/z ions reaching the detector at any particular moment in time [187,188]. This has resulted in an increase in proteogenomic depth of coverage and some increase in individual ORF coverage [189]. Meanwhile, in the MSi-TDP space, single molecule detection could solve problems related to proteoforms taking on multiple charge states and confounding analysis [190], but it is not clear how quantification is achieved. What is clear is that vendors and researchers are still largely focused on peptide-centric analysis, which perpetuates BUP yet also enhances the power of iTDP to deeply analyse proteoforms and thus proteomes.

With developments in instrumentation comes the need for development of software and algorithms to identify the peptides and proteoforms contained within MS/MS spectra. Data analysis in BUP has traditionally had a ‘flavour of the month’ mentality tempered by a software package’s ease of use and ease of integration into other analysis pipelines. MaxQuant [191] has long been the pipeline of choice because it is free, its output is readily accepted by pathway analysis pipelines [192], and it has a large community of users and resources available for those needing help. As computational resources have decreased in cost and developers have better understood how to make their software leverage those resources, BUP data analysis has moved to “Open Search” approaches in an attempt to assign more MS/MS spectra to a peptide sequence, especially those with PTM. Fragpipe [193] has been the most successful example of this, rapidly replacing MaxQuant as the pipeline of choice with downstream pathway pipelines being adapted to accept Fragpipe output [194].

The biggest change in BUP in the time of Proteomes’ existence has been the adoption of Data-Independent Acquisition (DIA) [195], which attempts to overcome the stochastic selection of peptides for fragmentation used in Data-Dependent Acquisition (DDA) approaches, which can result in missing values. DIA should result in a data file that contains fragmentation of every peptide able to be ionised, enabling retrospective analysis of those data files. However, DIA has also led to numerous analysis conundrums that were already a feature of BUP, including issues such as which peptide did a particular fragment ion belong to, thus simply deepening the protein inference problem. Ironically, successful DIA has been most reliant on comprehensive spectral libraries generated by DDA of highly fractionated peptide mixtures, although library-free methods such as that implemented in Progenesis [196] and DIA-NN [197] are in increasing use to overcome issues caused by Window-based DIA methods that do not measure intact peptide masses. Interestingly, library-free DIA has been a feature of Waters’ mass spectrometers that are able to separate peptides by ion mobility prior to measurement of their intact m/z and of the fragments [198]. Fragments with the same mobility value as the parent peptide must have come from that peptide, allowing a spectrum to be generated without confounding fragments from other peptides. Unfortunately, data from Waters instruments are incompatible with pipelines such as Fragpipe and DIA-NN due to the unwillingness of their developers to fully meet the needs of users, thus funnelling researchers to certain instrument vendors. Regarding proteoforms, DIA is limited by whether peptides defining a proteoform (e.g., by carrying sequence variants or PTM) are present in a spectral library; however, this is not a common characteristic of libraries because of the aforementioned lack of detection of these peptides in BUP, even with extensive fractionation. iTDP (2DE/LC/TMS) analysis of proteoforms could provide the reference TMS spectra required for comprehensive spectral libraries that include proteoform-specific peptides for DIA.

### 4.5. Improvements in the Depth of Proteome Analysis

The release of Bruker’s TIMSTOF platform also saw a resurgence in articles claiming to be performing ‘deep’ or ‘comprehensive’ proteome analysis [198,199,200,201]. As already emphasised, a lack of knowledge of the actual diversity of mature proteoforms in a cell makes claims of ‘deep’ proteome analysis rather pointless. ’Deep’ proteome analysis is thus an especially troubling term when applied to single cell proteomics (SCP) [130,202,203,204,205,206]. The current ‘State of the Art’ reports ~3000–5000 proteins (ORF products) able to be identified and quantified [130]; this recent study does not provide the necessary detail for critical evaluation (i.e., how many canonical proteins identified with a single peptide; how many with 2 or 3 or more), noting only a median protein sequence coverage of 12.9% for single cells. It is, of course, the variance in data that informs on quality. Furthermore, the need to analyse hundreds of cells on a single instrument is resulting in relatively short LC/MS analysis times (30 min or less) [130], which will lead to compromises due to scan speed limitations and ion suppression of co-eluting peptides which would otherwise be further separated in time. In addition, all of the aforestated issues with BUP still apply in SCP with reports not addressing the identification of proteoforms, although peptide speciation is beginning to be reported. For MSi-TDP, moving to single cells has necessitated the assessment of very large cells (i.e., muscle fibres) having only one or a few hyper-abundant species [207]. Again, this somehow seems to ‘justify’ the continued pursuit of BUP, but that simply leaves issues as they already exist and mires us in the promise of little advancement over ‘standardised’ proteogenomics over the next decade or more. There also appears to be little concern with sample preparation in regard to how the single cells are isolated and how that might affect the single cell proteome and thus how representative the findings are of the in situ native state. In this regard, while the issues associated with assessing cultured cell lines are more obvious, questions arise such as how does local heating during laser ablation [208] affect the proteomes of single cells close to and further from the line of excision? While it is not a question of how potentially important single cell analyses may be, much of the work has the quality of the technical attempts at tour de force studies [130] reminiscent of the first decade of BUP analyses and which still routinely appear in the literature [90]. But how much of that canonical cataloguing has effectively been turned into applicable knowledge? It is not what has been done that is important but rather what it means and thus what we learn from it. Perhaps the critical question should be ‘how genuinely and quantitatively deep can these analyses be pushed with current and developing technology’ rather than ‘how many canonical protein identifications can be quickly inferred with little if any knowledge of proteoforms’?

### 4.6. Developments in Alternative Proteome Analysis Technologies

There has always been a trend in science to adapt technologies from one field to another. Micro-arrays used for transcriptome analysis were adapted to protein arrays, although the technology did not achieve widespread use because of cost and other deficiencies [209,210]. Proteomics is currently seeing adoption of technology from genomics and NGS, where signal output is from amplification of oligonucleotide fragments attached to either antibodies (Olink) or aptamers (Somascan) that bind to a discreet part of a protein molecule, usually a series of amino acids (that could potentially include a PTM) [209,211,212,213]. While these technologies claim to be able to quantify up to 11,000 canonical proteins in a sample, they are subject to the same problem as protein arrays, that of being limited in their analysis to whatever “proteins” are targeted by the analysis panel. As we do not yet understand the diversity of proteoforms within an organism, it follows that these platforms also cannot measure specific proteoforms unless those are specifically targeted in the panel; thus, these analyses yield the same lack of proteoform detection and quantification as with BUP. This is not to say that these approaches have no value because there are reports of a >10 orders of magnitude dynamic range of detection, which is (minimally) estimated to be required for analysing samples such as serum or plasma. If the research question being asked genuinely fits within the limitations of these technologies, they may be effective tools to help uncover biology. However, as emphasised above, there is the need to subsequently pursue any finding to the proteoform level in order to identify the genuinely critical player(s).

The other emerging technology from genomics is nanopore sequencing, in which the passing of a linearised amino acid chain through a protein-based nanopore induces specific measurable electrical current changes depending on each particular amino acid [214,215]. There is recent evidence to suggest that PTM can also be identified and localised with this technology [216,217]. Issues to address include whether there are size limitations to the amino acid sequences that can be effectively linearised and ‘read’, and whether all (known) PTM can be distinguished or whether there will be overlapping and/or ambiguous signals for some.

Other technologies (e.g., refinements to Edman degradation, dendrimers, affinity matrices, BioID, DNA-PAINT, FRET X, CITE-seq and other RNA- and antibody-based techniques), in particular single molecule sequencing approaches, are in early stages but show promise, particularly when they demonstrably go beyond simply identifying canonical proteins but rather already address the need to analyse proteoforms [218]. Overall, there is potential in some of these approaches, but each has its own inherent technical limitations which are further complicated if proteoforms are the intended analytes for critical deep, *quantitative* analyses.

### 4.7. Integration with Other Omics Data

While proteoform-level analysis of the proteome is the necessary step forward that the field has to make, an equally important step is the integration of proteomics data with other omics data, especially metabolomics. Significant work has been done in this area with the creation of databases such as StringDB [219,220], PANTHER [221], and Reactome [222]; however, proteoform level data is lacking. Thus, although rarely done, researchers must consider and acknowledge the limitation that use of these software tools introduces in that this represents another form of inference since only canonical protein identifications are utilised. In their current state, obtaining usable data from these databases requires identified proteins to be submitted as gene names, as protein isoforms are not properly recognised. This is a curation problem that will only be solved by researchers submitting proteoform-defining information for review and inclusion and the database better recognising identifiers that define individual proteoforms (e.g., see suggested nomenclature [223]).

## 5. How to Move the Field More Rapidly Forward

Critically, broad recognition and exploitation of the complementarity of currently successful approaches is necessary, if not mandatory. Thus, for example, iTDP can quite effectively ‘fill the gaps’ until MSi-TDP has the necessary and sufficient capacity to fully address native proteomes across the full MW range of inherent proteoforms. Indeed, it has been known for over 20 years that TDP of larger species is best accomplished using an iTDP approach (at the time, in relation to MSi-TDP, inaptly named a ‘middle-down’ approach) [84]. Thus, by also utilising high-resolution 2DE as the front-end method to separate native proteoforms, current iTDP approaches most effectively capitalise on the complementary advances in both 2DE and BUP to enable genuinely deep, comprehensive proteome analyses. Rather than working largely in isolation from each other (if not actively against each other), which really has characterised much of proteomics over the last ~20+ years, the field must come to a transparent acceptance of the actual strengths and weakness of available methodologies and collaborate to capitalise on the former and in doing so, limit, if not eliminate, the latter. It is thus curious that MSi-TDP practitioners have avoided the obvious complementarity and improved analytical potential offered by 2DE. Furthermore, new technologies need to carefully continue through critical vetting processes that enable their ongoing evaluation as they are tested with increasingly complex samples and by several independent research teams in parallel. Such an approach will enable rapid addressing of issues rather than have them only become widely apparent years after full implementation of the method in the field. While that will certainly not provide an absolute guarantee that all problems will be identified in a timely fashion, it should significantly limit the now-usual pattern of having to address problems over the course of decades, which often leaves questionable data in the literature.

## 6. Consequences of a Failure to Address Proteoforms-the Price of Ignorance

The future lies in deeply understanding systems to identify specific and selective biomarkers and therapeutic targets. This is the only reasonable approach to genuine dissection of biological systems. Proteogenomics can not address the complexity of proteomes, although it might provide leads in those disorders having a direct genetic linkage; nonetheless, any potential leads must still then be pursued at the proteoform level. In this regard, for diseases resulting in abundance changes in a proteoform containing one or preferably more proteotypic peptides that can be subjected to targeted MS, this might provide a rapid diagnostic. But how often is this likely to be the case, noting that many of our most critical healthcare burdens are multi-factorial in nature? Furthermore, the current situation is that targeted MS studies (i.e., SRM/MRM; selected/multiple reaction monitoring) rarely target proteoforms beyond perhaps size variants (although exceptions are appearing [224]); appropriately using at least three peptides spanning the target species sequence is rarely done, and adding the complexity of specifically modified peptide standards to effectively calibrate the system for those PTM defining the proteoform of interest is a further demand and added expense. However, without these, specific proteoform identification and quantification is impossible.

Only a deep understanding of proteomes (and metabolomes, lipidomes, and transcriptomes) can provide the necessary functional and integrated understanding at the level of systems biology. The potential dangers of not deeply understanding the true functional components of systems—proteoforms—in an age in which techniques such as CRISPR are moving us ever closer to the realm of ‘routinely’ altering (defective) genes should be clear. How will a system that has developed without a specific functional protein/proteoform react to the expression of the ‘normal’ amino acid sequence? Will the system respond with (in)appropriate PTM? With any necessary PTM? The reality is that such treatments will (hopefully) target select cell types but these will reside within a whole system rather than the in vitro testing with specific cells in culture. We know already that, while generally effective, monoclonal antibody drugs are not entirely selective and thus even these therapeutics are not without side effects. How much more selective, and therefore perhaps devoid of off-target effects, would drugs be, regardless of their nature/type, if they were targeted to a specific (offending) proteoform—and thus that specific resulting folded (i.e., 3D) species—rather than broadly against a canonical amino acid sequence? This identifies another link in the chain that needs to be better addressed. While there has been substantial progress in now being able to predict protein structures using machine learning (‘AI’) approaches, these are not without serious issues [225]. In large part, this is likely still due to the influences of proteogenomics and the notion that all that is needed is a canonical ‘protein’ identification. What the structural prediction approaches realistically need to focus on is proteoforms [226]; only such an approach will prove truly useful to, for example, drug development [227]. Coming full circle, this again requires that proteoforms become the standard target level of all proteomic analyses if the field is to move constructively forward and make real contributions to health, agriculture, and environmental issues.

It is time to escape the technique or technology-centric biases that have dominated the field for far too long. These somewhat ego-driven, blindered approaches serve only the status quo and/or the development of new business approaches that still focus on proteogenomics, albeit with evolved technologies [130]. We must accept that ‘fitness-for-purpose’ applies, and must utilise and/or integrate available methods accordingly [7]. Clearly, development must always continue, as must routine (re)evaluation of established methods with the aim of constant improvement [10].

Among the critical questions that arise with available approaches, perhaps the most important is this: what if different ‘proteins’ are identified as important (e.g., significantly changing in abundance between test conditions) by iTDP vs. BUP because the latter does not discriminate proteoforms? What is missed? What is over-emphasized in importance? Which approach is the more relevant if we take genuine proteome complexity into account [11]?

## 7. Being the Difference: The Proteomes Journal Approach

At Proteomes, our established publication policies/expectations preferentially take the longer view, that native complexity must ultimately be the focus, and thus routinely and aggressively addressed where methods enable such critical analyses. It is thus expected that the concept of proteoforms and/or proteome complexity be at least touched upon in every published paper, even if the methods used do not directly enable proteoform assessment. Authors are expected to transparently address the pros and cons of their study, again with the understanding that the complexity of proteomes must be acknowledged and how the work contributes or will contribute to furthering that understanding.

What the discipline of Proteomics no longer needs is self-appointed leaders but rather leadership and vision with a focus on genuinely addressing the real complexity of proteomes. We must recognise this as the post-proteogenomic era. The question thus arises as to how to future-proof the field from (ongoing) approaches that largely address only the low-hanging fruit of canonical amino acid sequences or only low MW species? Proteomes believes it is time to spearhead a more complete working definition of proteomes and encourage innovative approach(es) to effectively drive the field forward as critically and quantitatively as possible. It is time to look forward and fully embrace the genuine complexity of proteomes and what it will mean to routinely analyse them as deeply and quantitatively as possible.

## 8. Conclusions/Directions/Rationale

Some openly bemoan the fact that it is (increasingly) difficult to continuously secure funding for ever-newer mass spectrometers (i.e., the ‘keeping-up-with-the-Joneses’ problem). Perhaps what funding agencies are/should be looking for is an effective analytical approach that provides the rigorous biological information necessary to understand and effectively target/dissect molecular mechanisms, and thereby identify rational new drug targets as well as biomarkers. Such rigorously identified therapeutics and biomarkers can and will subsequently survive appropriately rigorous validation, including clinical trials. Considering that ~86% of drug candidates between 2000–2015 failed in clinical trials, representing an exorbitant cost in both time and money [228], it is time to accept that ‘traditional’ approaches no longer suffice. Only the routine, deep proteoform level assessments provided by iTDP will yield critical systems biology knowledge. In this regard, it is surprising that LC/mass spectrometer manufacturers have not sought to offer more rigorous front-end analytical tools (i.e., 2DE and high-end imaging instrumentation) to best complement and capitalise on the LC/TMS equipment already marketed, and thus most effectively address the full needs of proteomics research [10]. Nonetheless, the future is promising, considering that rigorous iTDP approaches are well supported even by older (and less expensive) MS systems. That said, ion mobility MS may well prove to be a powerful tool in addressing proteoforms [229]. The ongoing development of nanopore approaches also appears promising in terms of potentially being capable of quantitatively assessing the full complement of proteoforms in a biological extract. While acknowledging this potential, enough questions remain, most specifically concerning proteoform complexity, that it seems unlikely that this approach will see wide-scale application to whole proteomes in the near immediate future, although one might imagine that quantitative targeted applications could appear at a reasonable pace. Direct elution of resolved proteoforms from 2D gel spots into a nanopore device might prove particularly advantageous. A critical focus on interactions—noting, however, that current widely used software applications, such as STRINGDB and PANTHER, address only canonical proteins and do not discern specific proteoform functions from the literature (if such information is even available)—will be essential to our understanding of systems at a genuinely functional level. This, then, also emphasises the need for better (1) structural analyses and predictions, that focus on proteoforms rather than only canonical amino acid sequences; (2) spatial resolution (e.g., in MS imaging); (3) temporal resolution (e.g., to assess transient proteoforms, perhaps even in signalling networks) [230]; (4) understanding the implications of PTM crosstalk [231]; and careful consideration of (4) the potential applications of machine learning to addressing data analysis and interpretation [232]. With such routine deep proteome analyses comes the very real promise of far more selective biomarkers, drug targets, and personalised—or even realistically individualised—medicine. Only a rigorous focus on analytical quality will get us there.

## Data Availability

Not applicable.

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
