# Peer review of "Proteomics—The State of the Field: The Definition and Analysis of Proteomes Should Be Based in Reality, Not Convenience"

_proteomes, 2024, doi:10.3390/proteomes12020014_

Round 1
Reviewer 1 Report
Comments and Suggestions for Authors
This paper provides an excellent view at current proteomics research with all its flaws. I am looking forward to see it published to be able to refer to it in my teaching.
Being not a native speaker, I found the reading difficult, however. This is not a criticism, but rather feedback. The use of the English language is so advanced in the manuscript – one sentence covering 7 lines and more with multiple separation signs – that I had to read some text several times and still had difficulties to catch its meaning. We were taught in school that English is the language of short sentences and, for the foreigner, this certainly helps easier reading. On the other hand, I have enjoyed to learn more about other writing styles. I particularly liked the references to Einstein and Kennedy (I tried to quote Sherlock Holmes one time in a scientific paper and was laughed at).
As for the scientific content – I am 100% behind it, in fact, I was reminded of a few facts which I had not been properly aware of.
Minor: 2 typos: p11 l552 and p13 l644: “Furthermore, “ (comma, no dot)
Author Response
As the Reviewer has quite reasonably noted, we have sometimes allowed the complexity of proteomes to be reflected in the complexity of our sentences. We have taken these comments to heart and sought to trim/rewrite some particularly noisome sentences where possible. We hope this is to the Reviewer's satisfaction.
Reviewer 2 Report
Comments and Suggestions for Authors
The manuscript is well written and documented.
I found quite difficult to follow the introduction, since it appears to me too hard. I recommend including a section for acronyms, where the non-specialized reader can revisit to consult different technical terms.
In addition, a section summarizing the most relevant results using well conducted techniques in proteomics in different disciplines would result very illustrative and interesting for Proteomes readers.
Please, move the title of the section “Conclusions/Direction/Rationale” to the next page to avoid title disconnection from the section text.
Author Response
As the Reviewer has quite reasonably noted, we have sometimes allowed the complexity of proteomes to be reflected in the complexity of our sentences. We have taken these comments to heart and sought to trim/rewrite some particularly noisome sentences where possible. We hope this is to the Reviewer's satisfaction. We believe that the requested section is covered in the main body of the manuscript and that summarising the relevant results moves the manuscript from a Perspective to a Review and there are many Reviews (cited in this manuscript) that provide the information requested.
Reviewer 3 Report
Comments and Suggestions for Authors
The paper presents a very inspiring critical review o the status of what is now called proteomics. The questions clearly answer the main problem with the discipline, the technical approach leading to blackboxes allowing to use the fdast and easy way. This is even more pronounced when moving to the "single cell" concept.
The revisit of 2D or at least intact proteins separation methods is appropriate, as the use of high accuracy, high resolution MS instruments. The strategy should be pathway driven more than the comprehensive "proteome" approach.
The top down approach may solve part of the problem if focused on a proteins "family"and could have been reviewed more tha cited in the introduction.
The paper should maybe contain a paragraph on how to cope with the dynamic aspects of the biological systems. But this is maybe the next step.
Author Response
We thanks the reviewer for their comments. Measuring the dynamic aspects of biological systems is an entire review in itself and we can't cover it in this Perspective.
Reviewer 4 Report
Comments and Suggestions for Authors
The authors did a trully extensive work to describe their opinions on the future of proteomics analysis. They focus on the importance of the field taking a breath and have a look where it stands.
According to the authors, we are in the post-proteogenomics era and we need more biological relevance on what is detected and measured by proteomics analyses. A focus in proteoforms and their accurate separation and detection is important.
Then the authors express their views on the fields and suggest specifically the steps that are required to provide more biological relevance to the proteomics profiles and how important that would be even for AI modeling and biomarker evaluation.
Although, some parts can be debated (i.e DIA caused more issues than expected), it is their honest opinion and perspectives like this need to be published because they raise important topics and concerns. I would suggests a table summarising the improvements they suggest and, perhaps, a simple graphical representation of their vision.
Author Response
We thanks the reviewer for their comments. A table summarising the improvements we suggest would just give readers a reason not to read the broader arguments that we present. This we won't be providing one.